# Middle Eastern Genetic Variation Improves Clinical Annotation of the Human Genome

**DOI:** 10.3390/jpm12030423

**Published:** 2022-03-09

**Authors:** Sathishkumar Ramaswamy, Ruchi Jain, Maha El Naofal, Nour Halabi, Sawsan Yaslam, Alan Taylor, Ahmad Abou Tayoun

**Affiliations:** 1Al Jalila Genomics Center, Al Jalila Children’s Hospital, Dubai, United Arab Emirates; sathish.kumar@ajch.ae (S.R.); ruchi.jain@ajch.ae (R.J.); maha.elnoufel@ajch.ae (M.E.N.); nour.halabi@ajch.ae (N.H.); sawsan.yaslam@ajch.ae (S.Y.); alan.taylor@ajch.ae (A.T.); 2Center for Genomic Discovery, Mohammed Bin Rashid University of Medicine and Health Sciences, Dubai, United Arab Emirates

**Keywords:** Middle East Variants, whole exome sequencing, whole genome sequencing, knockouts, common variants

## Abstract

Genetic variation in populations of Middle Eastern origin remains highly underrepresented in most comprehensive genomic databases. This underrepresentation hampers the functional annotation of the human genome and challenges accurate clinical variant interpretation. To highlight the importance of capturing genetic variation in the Middle East, we aggregated whole exome and genome sequencing data from 2116 individuals in the Middle East and established the Middle East Variation (MEV) database. Of the high-impact coding (missense and loss of function) variants in this database, 53% were absent from the most comprehensive Genome Aggregation Database (gnomAD), thus representing a unique Middle Eastern variation dataset which might directly impact clinical variant interpretation. We highlight 39 variants with minor allele frequency >1% in the MEV database that were previously reported as rare disease variants in ClinVar and the Human Gene Mutation Database (HGMD). Furthermore, the MEV database consisted of 281 putative homozygous loss of function (LoF) variants, or complete knockouts, of which 31.7% (89/281) were absent from gnomAD. This set represents either complete knockouts of 83 unique genes in reportedly healthy individuals, with implications regarding disease penetrance and expressivity, or might affect dispensable exons, thus refining the clinical annotation of those regions. Intriguingly, 24 of those genes have several clinically significant variants reported in ClinVar and/or HGMD. Our study shows that genetic variation in the Middle East improves functional annotation and clinical interpretation of the genome and emphasizes the need for expanding sequencing studies in the Middle East and other underrepresented populations.

## 1. Introduction

Cataloguing human genetic variation at an unprecedented scale has significantly improved the clinical interpretation of genetic variants found in patients with Mendelian disorders [1]. The 1000 Genomes Project created a catalogue of human genetic variations applying whole-exome sequencing (WES) and whole-genome sequencing (WGS) on 2504 individuals from 26 different populations [2]. This project characterized over 88 million variants in the human genome, including >99% of single nucleotide variants (SNVs), with a frequency of >1% for a variety of ancestries. This was a valuable resource for research involving the genetic basis of human disorders. Besides, WES of 6515 individuals from European American and African American populations was performed by the NHLBI GO Exome Sequencing Project (ESP) to assess the distribution of mutation ages and predicted that about 86% of SNVs had recent origins [3]. Large-scale reference data sets established by the Exome Aggregation Consortium (ExAC) [4], aggregating 60,706 exome sequences, provided a more comprehensive summary of human genome variations; later, the Genome Aggregation Database (gnomAD) aggregated 125,748 exome sequences in addition to 15,708 whole-genome sequences of unrelated individuals from various ancestries. These publicly available datasets are beneficial for use by the clinical and scientific community. However, current genomic databases still fall short of capturing the full representation of human genomic diversity. For example, the Middle Eastern and African populations, among others, remain highly underrepresented in the Genome Aggregation Database (gnomAD), which is the most comprehensive compendium of human genetic variations to date [1,5,6]. This lack of representation is a missed opportunity to fully understand the human genome and to functionally and clinically annotate its variation. 

The Middle Eastern population, spanning North Africa, the Arabian Peninsula, and the Syrian desert, has a long history of admixture and migration leading to a rich and highly diverse genetic architecture. In addition, this population is characterized by significant endogamy, relatively high consanguinity rates, extended family structures, and an advanced paternal and/or maternal age at conception [7]. As a result, a high prevalence of Mendelian recessive disorders is expected [8,9], given the higher burden of regions of homozygosity (ROH) in this population [5]. Furthermore, these extended ROH regions can be enriched for biallelic gene knockouts in apparently healthy individuals, shedding light on the biological roles of several genes, and empowering the clinical interpretation of the genome. 

Expanding sequencing studies in the Middle East would, therefore, be undoubtedly a unique opportunity for advancing the human genetics field. However, few attempts have been made [7,10] to characterize the genetic variations in the Middle East population, while the impact of cataloguing this variation, albeit on a small scale, on the clinical interpretation of genetic variants remains to be elucidated. 

In the present study, we have assembled sequencing data from Qatar [10] and the Greater Middle East (GME) [7] to highlight the contribution of variants from this population to existing and commonly utilized genomic variation datasets, specifically gnomAD. We also capture disease pathogenicity assertions of rare (based on gnomAD) variants in the Human Gene Mutation (HGMD) [11] and ClinVar databases [12], which we annotate with allele frequency in the Middle East cohort. These comprehensive variant sets comprise putative common Middle East disease variants and add a unique set of gene knockouts. Furthermore, our analysis questions the pathogenicity of previously reported disease variants that might be putative polymorphisms. This study demonstrates the importance of capturing genetic variation in the Middle East and highlights the integration of different variant datasets to improve the clinical annotation of the human genome. 

## 2. Materials and Methods

### 2.1. Study Cohort

We compiled sequencing data from 1005 individuals from Qatar (88 whole genomes and 917 whole exomes) [10] and 1111 healthy individuals from The Greater Middle East (GME) exome sequencing study to characterize variation in the Middle East (Figure 1A). Sequencing protocols and variant calling pipelines are detailed in the original studies [7,10]. The quality control metrics are summarized in Appendix A.

Individuals from Qatar were either Bedouin (*n* = 490), Arabs (*n* = 193), Persian (*n* = 170), South Asian (*n* = 76), Sub-Saharan African (*n* = 70), European (*n* = 5), or African Pygmy (*n* = 1). On the other hand, individuals from the GME dataset were from Northeast Africa (NEA, *n* = 423), Northwest Africa (NWA, *n* = 85), the Arabian Peninsula (AP, *n* = 214), the Turkish Peninsula (TP, *n* = 140), the Syrian Desert (SD, *n* = 81), and Persia and Pakistan (PP, *n* = 168) (Figure 1B,C).

### 2.2. Middle East Variation (MEV) Database

Data from both the GME and Qatar studies were processed using the GATK workflow in accordance with best practices, including the elimination of duplicate reads, aligning pair-end reads to the human reference genome NCBI Build 37 using BWA (version 0.7.5). To address batch effects in both datasets, authors of the GME and Qatar studies have carried out extensive batch adjustments that produced comparable results across centers followed by different statistical models and filters to reduce sequencing artifacts and assure high-quality variants. In the GME study, principal component analysis (PCA) was carried out apart from standard filtering criteria on the set of variants to identify potential batch effects between sequencing labs; then, sequencing artifacts were observed and eliminated from the data. On the other hand, in the Qatar study, sequencing was performed at three different centers. To control batch effects, authors filtered variants below a threshold depth *d* and threshold variant allele count *v*, such that the mean novel SNP rate was consistent across batches within genomic intervals covered by the intersection of all batches [7,10]. These high-quality variants from both Qatar and GME datasets (VCF/TSV files) were obtained and then merged using hg19 chromosomal coordinates via an in-house pipeline to generate a non-redundant (unique variant locus and alternate alleles) Middle East Variation (MEV) database, which is available upon request (Table 1). 

### 2.3. Functional and Clinical Annotation of Variants in the Middle East Variation (MEV) Database

Merged MEV variants were then annotated with gnomAD allele frequency using the gnomAD genomes V3 exonic regions merged with exomes V2.1.1 dataset, which has 29,812,147 variants from unrelated individuals sequenced as part of various disease-specific and population genetic studies [1]. Variants were then further annotated using multiple public and commercial databases, including HGMD v2021.3 [11], ClinVar v29032021 [12], NCBI-RefSeq-105 [13], OMIM [14], and the UniProt database [15] using an in-house pipeline.

Our variant annotation pipeline overlays variant positions with extensive resources from the NCBI-RefSeq-105 database and assigns variant consequences [16] with respect to each transcript and protein within the NCBI-RefSeq-105 database. Additional gene annotations, such as disease phenotype, pathogenicity, and function, were obtained from various data sources [11,12,14,15]. We have not applied any criteria to predict the pathogenicity of variants. High-impact coding variants (missense (excluding synonymous), stop lost/gain, splice acceptor/donor (±1, 2), frameshift) in the MEV database were then classified as “unique” or “reported” if they were absent from or reported at least one time in gnomAD 2.1.1, respectively (Table 1).

Annotated variants were subsequently classified into two main classes, I and II, as shown below. 

### 2.4. Class I: Common Middle East Disease Variants (CMEDVs)

To obtain this list, heterozygous variants with minor allele frequency (MAF) > 1% in our MEV database were intersected with rare (<1% total allele frequency in gnomAD) variants with disease mutation (DM) status in or with pathogenic (P) and likely pathogenic (LP) classifications and ≥1 star in ClinVar. HGMD-DM variants with benign and/or likely benign classifications in ClinVar were excluded from this list.

### 2.5. Class II: Putative Knockouts (KOs) 

Knockouts (nonsense, frameshift, and ±1, 2 splice site) in the MEV database were filtered for further manual curation. High-confidence LoF status was extracted from gnomAD 2.1.1 for LoF variants that were present in this database. To infer high-confidence LoF impact for LoF variants absent from gnomAD 2.1.1, we excluded variants affecting initiator codons or those located in the last coding exon (CDS) or within 50 bp of the penultimate CDS. Variants located in alternatively spliced exons were excluded if the exon was not functionally or clinically relevant based on clinically curated transcripts in disease databases and/or expression data in the Genotype-Tissue Expression (GTEx) dataset [17]. In addition, we removed LoF variants with the low quality associated with high homology regions or pseudogenes, as described [18]. High-confidence LoFs were manually verified using the Alamut program v2.11. 

## 3. Results

### 3.1. Middle East Variation (MEV) Database

A total of 26,228,226 non-redundant variants (see methods) were merged from the GME and Qatar datasets to establish the MEV database. We focus on the set of high-impact coding (missense, stop gain/loss, splice acceptor/donor (±1, 2), frameshift) variants (*n* = 600,987) affecting RefSeq transcripts/exons (Methods), given such variants represent the majority of disease variants [11]. Of those, 318,242 (53%) variants were absent from gnomAD 2.1.1 exomes (Table 1) representing unique coding variation in this Middle Eastern cohort. There were slightly more singleton unique coding variants (41.37%) compared to the reported ones (38.1%) (Appendix A).

### 3.2. Common Middle East Disease Variants (CMEDVs)

Of the total HGMD-DM and ClinVar P/LP variants that were relatively rare in gnomAD (MAF < 1%), 3480 were observed in the MEV database and 39 of those variants were common (MAF > 1%) or had at least 1 homozygote in the MEV database (Appendix A and Figure 2a). Those common Middle East disease variants (CMEDVs), which were mostly missense (*n* = 37, 94.8%), affected 37 genes with P/LP assertions in ClinVar (*n* = 1) or DM status in HGMD (*n* = 38) (Figure 2b).

While it is highly likely that a significant proportion of those 39 variants can be reclassified to benign or likely benign based on the MEV allele frequency, it is also possible that a subset can still be clinically significant. In fact, two ClinVar variants had at least one star with no conflicting LP interpretations and might be common founder mutations in the Middle East, associated with primary ciliary dyskinesia (*DNAAF4*) and 3-methylcrotonyl-CoA carboxylase 2 deficiency (|*MCCC2*) (Appendix A), leading to a higher incidence of such diseases in this region. 

### 3.3. Knockouts in the MEV Database 

There were 281 knockouts in the MEV database and 89 of those (31.7%) were not present in the homozygous state in gnomAD 2.1.1 exomes and were of high quality in reportedly healthy individuals in the GME dataset, thus representing putative unique Middle Eastern high confidence knockouts (Appendix A and Figure 3a). This unique homozygous variant set was mostly frameshift (*n* = 52, 58.4%) (Figure 3b), impacting 83 genes where at least 42 of those genes had some disease association in OMIM databases (Figure 3c). 

Of the 89 unique putative knockouts, which were of high quality identified in reportedly healthy individuals in the GME dataset, 24 genes, in particular, had several DM and P/LP reported in HGMD and ClinVar, respectively (Appendix A). Examples include *SPG11* (MIM# 610844) associated with autosomal recessive spastic paraplegia, *RAB3GAP2* (MIM# 609275) linked to autosomal recessive Marsolf syndrome, and *NPHP4* (MIM# 607215) linked to autosomal recessive nephronophthisis. While many of these can be true knockouts with implications for disease penetrance and expressivity, it is also possible that the exons impacted by those homozygous LoF variants might not be clinically relevant and should be excluded from curated transcripts for clinical annotation. In fact, human transcriptomic data in the Genotype-Tissue Expression (GTEx) database showed that 11 out of the 27 (41%) affected exons have relatively low expression levels (proportion expressed across transcripts, pext score < 0.5) (Appendix A).

## 4. Discussion

We have aggregated the largest variant database from 2116 individuals of Middle Eastern origin and characterized the impact of this dataset on the functional and clinical annotation of the human genome. We, therefore, focus on coding variants that represent the majority of reported disease variants to date [11] and show that 53% of those variants in the MEV database are absent from gnomAD 2.1.1 exomes and are thus specific to the Middle East population. 

Using the MEV database, we highlight 39 variants, which were previously reported as rare clinically significant variants in disease databases (ClinVar and HGMD), yet were common (MAF > 1%) or present in the homozygous state at least once in Middle Eastern individuals. While this information might question the pathogenicity of a proportion of those variants, specifically some of the HGMD-DM variants (*n* = 38) and those with conflicting interpretations in ClinVar (*n* = 22), others might be clinically significant founder mutations, as shown above. 

Our MEV database also consists of 281 high-confidence homozygous LoF variants, the majority of which (89/281, 31.7%) were absent from gnomAD exomes 2.1.1. This unique set affects 83 genes, of which 24 had several clinically significant variants reported in ClinVar and HGMD yet were identified in reportedly healthy individuals in our dataset. While it might question the clinical validity of some of the impacted genes, this information might, in fact, refine our understanding of penetrance, expressivity, and severity for the diseases caused by those genes. Finally, it is also possible that the current exon and transcript structure and expression for some genes should be revisited in light of this information (see Section 3).

Similarly, Fattahi and his colleagues performed whole-exome sequencing on 800 individuals from eight major Iranian ethnic groups and identified 1,575,702 variants, of which 308,311 were novel (19.6%), compared to current databases, including gnomAD [19]. More recently, a study sequencing whole genomes from 6218 Qatar individuals identified 74,783,226 variants, of which 28% were not present in current databases, mainly 1KG project, Human Origin dataset, and GME [20]. These studies and our analyses highlight the importance of expanding genomic sequencing studies among diverse underrepresented populations, which include variation that has not yet been sampled. This will subsequently enhance our understating of human genetic variation along with its biological and clinical effects. 

Our study is limited by its size (*n* = 2116 individuals), which might not capture the full genetic diversity in the Middle East. Our results suggest that this unique MEV variant dataset improves the clinical and functional annotation of the human genome. Despite its small size, however, the value of cataloguing genetic variation in this population, as demonstrated in this study, should encourage the expansion of sequencing studies in the Middle East and other underrepresented populations, to maximize our understanding of the human genome.

## Figures and Tables

**Figure 1 jpm-12-00423-f001:**
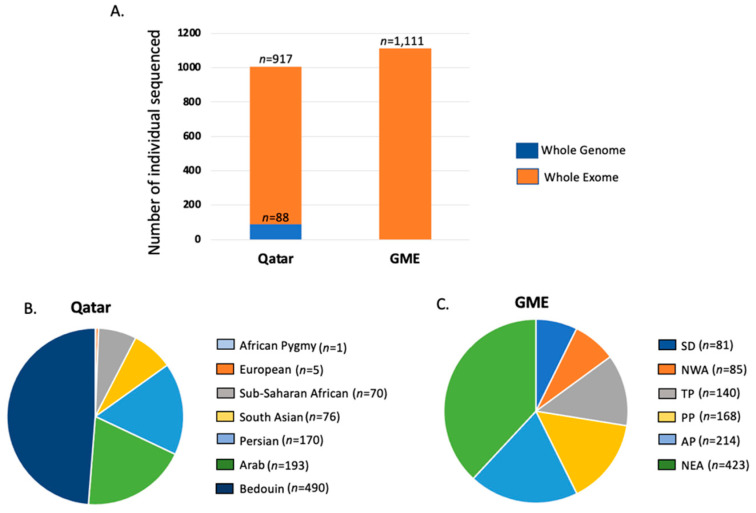
Samples used for this study. (**A**) Data from a total of 88 whole genomes and 2028 whole exomes from the Qatar and Greater Middle East (GME) studies were aggregated in this study; (**B**) ancestry distribution of samples from the Qatar dataset; (**C**) ancestry distribution of samples from the GME dataset. NWA, Northwest Africa; NEA, Northeast Africa; TP, Turkish Peninsula; SD, Syrian Desert; AP, Arabian Peninsula; PP, Persia and Pakistan.

**Figure 2 jpm-12-00423-f002:**
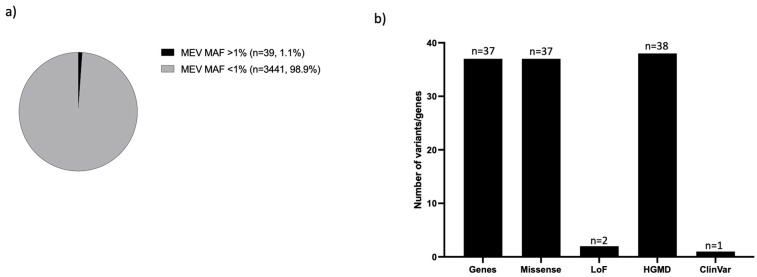
Characterization of common Middle East disease variants (CMEDVs). (**a**) Percentage of CMDEVs (MEV MAF > 1%) and rare (MEV MAF < 1%) set represents variants DM, or P/LP, which are also rare (<1%), in gnomAD. (**b**) Effect of CMEDVs and total number of genes impacted by those variants and distribution of CMEDVs, which are reported at different star levels in ClinVar and HGMD. Number of variants = Number of variants that are Missense or LoFs, or are in HGMD and ClinVar. Number of genes = Number of genes in CMEDV.

**Figure 3 jpm-12-00423-f003:**
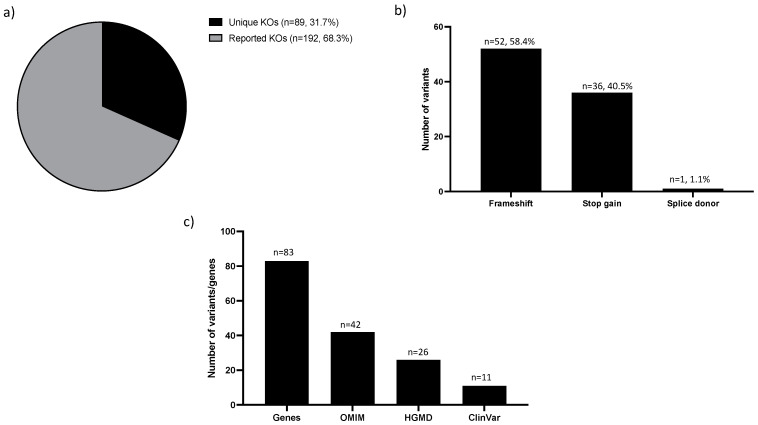
Characterization of high confidence knockouts (KOs) in the MEV database. (**a**) Distribution of unique (present in MEV database only) and reported (present in both MEV database and gnomAD) knockouts. (**b**) Effects of unique KOs variants. Number of variants = Number of stops gained, frameshift, splice acceptor, stop lost, and splice donor variants. (**c**) Distribution of unique KOs genes in different disease databases (ClinVar, HGMD, and OMIM). Number of variants = Number of variants in OMIM, CLinVar, and HGMD. Number of genes = Number of genes in Unique KOs.

**Table 1 jpm-12-00423-t001:** Distribution of variants in MEV database.

	Total Variants	SNPs	Indels
Total MEVs	26,228,226	21,180,218	5,048,008
Total coding variants	600,987	534,287	66,700
Unique coding variants *	318,242 (53%)	263,680	54,562
Reported coding variants **	282,745 (47%)	270,607	12,138

* Unique coding variants = Variants not reported in gnomAD 2.1.1. ** Reported coding variants = Variants reported at least once in gnomAD 2.1.1.

## Data Availability

Not applicable.

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
