# Peer review of "Middle Eastern Genetic Variation Improves Clinical Annotation of the Human Genome"

_jpm, 2022, doi:10.3390/jpm12030423_

Round 1

Reviewer 1 Report

In the resubmitted manuscript entitled "Middle Eastern Genetic Variation Improves Clinical Annotation of the Human Genome", by Sathishkumar Ramaswamy and colleagues authors aggregate genomic variants obtained from two large sequencing studies involving 2116 individuals from Qatar and Greater Middle East region, in order to improve clinical variant interpretation and underpin the importance of expanding current databases of genomic variation in the population with individuals of Middle Eastern ancestry. 

Compared to previously submitted versions authors have made visible improvements in the text quality and data presentation. However, some of the major concerns from the last revision remained unanswered. 

1) Methods, lines 112-115 - Described data processing do not reduce batch effects. As authors wrote, they represent best practices suggested by Broad Institute with the aim to get the best possible results, but, with exception of duplicate removal, none of them can eliminate artefacts induced by sample preparation or sequencing chemistry. Autors should modify this sentence accordingly. 

2) Results in table 1 indicate somewhere higher abundance of indels in the MEV database compared to reported from other populations (around 19% of MEV VS 10-15% in other populations; https://doi.org/10.1038/nature15393). Bearing in mind that indels and nonsense mutations are very rare in the coding regions (due to their natural negative selection), and the fact that the variants in MEV database are largely obtained through whole exome sequencing experiments, their high abundance in this cohort may indicate presence of possible sequencing artefacts. 

3) In line with previous comments, and as noted in my previous review, in the supplementary table 1 authors show lower PHRED-scaled filtering threshold in the GME study compared to Qatari, which indicates 10x higher chance that the variants called in the GME study are incorrect (error rate 1% in GME and 0.1% in the study from Qatar). Unfortunately, authors did not try to tackle this problem in the current version of the manuscript. Bearing in mind conclusions of this study and prevalence of disease causing variants in the healthy MEV population authors need to put greater effort in proving that the observed variants are not artefacts from the GME study. For example, authors can examine where these disease causing variants are located in the human genome (e.g., are these inside homopolymer regions), are these mutations occurring in only a subset of individuals, are majority of them coming from one of the two studies, is there any indication that some of the individuals from healthy population were actually diseased? Furthermore, adding more information about this aspect in the method and result section would greatly aid in convincing reviewers about validity of the study. 

4) Results, lines 163-166 - Authors should explain which criteria they used to select high impact missense variants.

5) Authors should discuss their findings in the context of what was reported in other studies. Current discussion is rather short and largely repeats results and findings of the current study.

Minor comments:

1) Supplementary Table 1 – data presented in the table does not show variant quality metrics but rather filtering and raw data processing strategy. Please correct accordingly.

Author Response

In the resubmitted manuscript entitled "Middle Eastern Genetic Variation Improves Clinical Annotation of the Human Genome", by Sathishkumar Ramaswamy and colleagues authors aggregate genomic variants obtained from two large sequencing studies involving 2116 individuals from Qatar and Greater Middle East region, in order to improve clinical variant interpretation and underpin the importance of expanding current databases of genomic variation in the population with individuals of Middle Eastern ancestry. 

Compared to previously submitted versions authors have made visible improvements in the text quality and data presentation. However, some of the major concerns from the last revision remained unanswered. 

  • Methods, lines 112-115 - Described data processing do not reduce batch effects. As authors wrote, they represent best practices suggested by Broad Institute with the aim to get the best possible results, but, with exception of duplicate removal, none of them can eliminate artefacts induced by sample preparation or sequencing chemistry. Autors should modify this sentence accordingly. 

Thank you for your comment we have modified this sentence.

  • Results in table 1 indicate somewhere higher abundance of indels in the MEV database compared to reported from other populations (around 19% of MEV VS 10-15% in other populations; https://doi.org/10.1038/nature15393). Bearing in mind that indels and nonsense mutations are very rare in the coding regions (due to their natural negative selection), and the fact that the variants in MEV database are largely obtained through whole exome sequencing experiments, their high abundance in this cohort may indicate presence of possible sequencing artefacts. 

Thank you for pointing this out. Authors from Qatar and GME have actually carried out extensive statistical methods, principal component analysis and implemented several filters to remove sequencing artifacts due to batch effects. We have used data downstream of these analyses in this manuscript. Therefore, we assume that sequencing artifacts should at least be minimal in our dataset.  

We have modified methods section to highlight the analyses performed in the GME and Qatar studies regarding the sequencing artifacts issue.

  • In line with previous comments, and as noted in my previous review, in the supplementary table 1 authors show lower PHRED-scaled filtering threshold in the GME study compared to Qatari, which indicates 10x higher chance that the variants called in the GME study are incorrect (error rate 1% in GME and 0.1% in the study from Qatar). Unfortunately, authors did not try to tackle this problem in the current version of the manuscript. Bearing in mind conclusions of this study and prevalence of disease causing variants in the healthy MEV population authors need to put greater effort in proving that the observed variants are not artefacts from the GME study. For example, authors can examine where these disease causing variants are located in the human genome (e.g., are these inside homopolymer regions), are these mutations occurring in only a subset of individuals, are majority of them coming from one of the two studies, is there any indication that some of the individuals from healthy population were actually diseased? Furthermore, adding more information about this aspect in the method and result section would greatly aid in convincing reviewers about validity of the study. 

Thank you for your suggestion. We have checked the genomic context of all knockout locations (homopolymer or low complexity region) and updated supplementary tables (2 & 3) with this information. As can be seen, only one location was labeled as “low complexity”. Please note that all homozygous calls were of high quality (Filter=PASS) in the GME browser (http://igm.ucsd.edu/gme/data-browser.php). In addition, the GME study reported that all included individuals were healthy. Finally, we agree with the reviewer that such knockouts are likely deleterious and should be rare or not observed at all (in fact, each one of the homozygous variants are seen only once in one individual). While their occurrence can be attributed to sequencing artifacts due to batch effects, another reason could be that those variants are located in dispensable exons. In fact, our exon-level expression analysis show that that 41% of affected exons in disease genes had relatively low expression levels based on the Genotype Tissue Expression (GTEx) database. 

  • Results, lines 163-166 - Authors should explain which criteria they used to select high impact missense variants.

We used the term “high impact” variants to refer to those altering the coding sequence (missense, nonsense, frameshift, splice ½). We have not applied any criteria to predict impact of missense variants.

  • Authors should discuss their findings in the context of what was reported in other studies. Current discussion is rather short and largely repeats results and findings of the current study.

Thank you for your comment, we have summarized other genome studies carried out in this region.

Minor comments:

1) Supplementary Table 1 – data presented in the table does not show variant quality metrics but rather filtering and raw data processing strategy. Please correct accordingly.

Thank you for your comment, we have corrected as suggested.

Reviewer 2 Report

Genomics structure of populations in Middle Eastern area is important to implement precision medicine, including rare disease diagnose, pharmacogenomics, in this area. Most databases and research focus on European populations. A good genomics database, detailed analysis and clinical applications for Middle Eastern populations are also provide an example to other underrepresented populations.

In this article, authors established a Middle Eastern genomics dataset by aggregating original data from Qatar and     GME projects. Comprehensive annotating is conducted on the dataset. Analysis results shown that unique variants in Middle Eastern populations are critical to disease diagnoses.

As suggested by the authors, larger cohort is required to build a high-quality genomics database of middle eastern populations. As I known, lots of middle eastern medical and genomics institutions are working on whole genome sequencing based national genomics projects. The authors can build a better MEV database for clinical applications in near future.

Some suggestions:

  1. As I know, besides Qatar genome and GME project described in the article, Saudi Genome project had published results. I think these works should be added into the introduction.
  2. Statistical analysis has been conducted in this article. It is very interesting to see how the MEV database effect clinical applications of genomics technologies in middle eastern population. For example, how does it improve diagnose quality of rare diseases?
  3. More annotations, such as pharmacogenomics, chronic diseases, GWAS, can be added to analysis.

Author Response

Genomics structure of populations in Middle Eastern area is important to implement precision medicine, including rare disease diagnose, pharmacogenomics, in this area. Most databases and research focus on European populations. A good genomics database, detailed analysis and clinical applications for Middle Eastern populations are also provide an example to other underrepresented populations.

In this article, authors established a Middle Eastern genomics dataset by aggregating original data from Qatar and     GME projects. Comprehensive annotating is conducted on the dataset. Analysis results shown that unique variants in Middle Eastern populations are critical to disease diagnoses.

As suggested by the authors, larger cohort is required to build a high-quality genomics database of middle eastern populations. As I known, lots of middle eastern medical and genomics institutions are working on whole genome sequencing based national genomics projects. The authors can build a better MEV database for clinical applications in near future.

Some suggestions:

  1. As I know, besides Qatar genome and GME project described in the article, Saudi Genome project had published results. I think these works should be added into the introduction.

Thank you for your suggestion, we don’t have access to the Saudi genome project.

We are in the process of building Emirati population specific exome database, hopefully we will aggregate this data in MEV database in near future.

  1. Statistical analysis has been conducted in this article. It is very interesting to see how the MEV database effect clinical applications of genomics technologies in middle eastern population. For example, how does it improve diagnose quality of rare diseases?

Thank you for your comment, the value of cataloguing genetic variation in this population, as demonstrated in this study, should encourage the expansion of sequencing studies in the Middle East and other underrepresented populations, to maximize our understanding of the human genome. this information might in fact refine our understanding of penetrance, expressivity, and severity for the diseases caused by those genes.

  1. More annotations, such as pharmacogenomics, chronic diseases, GWAS, can be added to analysis.

Thank you for your suggestion, In the present study, we have included tissue specific gene expression (GTX), genetic phenotype (OMIM), in addition HGMD annotation which suggest the likelihood of variants associated with disease phenotypes.

Round 2

Reviewer 1 Report

In the revised manuscript, authors addressed most of my major concerns, including the ones related to the sequencing artefacts. 

My remaining major concern remains definition of the high impact variants, where authors also included missense variants without prior filtering for pathogenicity. Defining all missense variants as high impact and putting them in the same group with nonsense and frameshift suggests to the reader that some sort of selection for pathogenic variant was performed. Furthermore, missense variants in this dataset may include synonymous variants, and for these, vast majority will not have any impact. Indeed, in the supplementary table 2 most of missense variants are annotate as conflicting or with uncertain significance. Authors should ascertain that none of these missense variants is synonymous (since they are unlikely to have any major impact) and explicitly mention in the method section that no further selection for pathogenic mutations was performed.

Author Response

Thank you for pointing this out. In fact, none of the missense variants were synonymous. We have reframed the methods section as requested. 

Our variant annotation pipeline overlays variant positions with extensive resources from the NCBI-RefSeq-105 database and assigns variant consequences [18] with respect to each transcript and protein within the NCBI-RefSeq-105 database. Additional gene annotation such as disease phenotype, pathogenicity and function were obtained from various data sources [14], [11], [12], [17]. We have not applied any criteria to predict the pathogenicity of  variants. High impact coding variants (missense (excluding synonymous), stop lost/gain, splice acceptor/donor (±1,2), frameshift) in MEV database were then classified as “unique” or “reported” if they were, respectively, absent from or reported at least one time in gnomAD 2.1.1 (Table 1).

Round 3

Reviewer 1 Report

The authors have satisfactorily addressed my concerns.